# Tissue-Oxygen-Adaptation of Bone Marrow-Derived Mesenchymal Stromal Cells Enhances Their Immunomodulatory and Pro-Angiogenic Capacity, Resulting in Accelerated Healing of Chemical Burns

**DOI:** 10.3390/ijms24044102

**Published:** 2023-02-17

**Authors:** Marina V. Volkova, Ningfei Shen, Anna Polyanskaya, Xiaoli Qi, Valery V. Boyarintsev, Elena V. Kovaleva, Alexander V. Trofimenko, Gleb I. Filkov, Alexandre V. Mezentsev, Sergey P. Rybalkin, Mikhail O. Durymanov

**Affiliations:** 1School of Biological and Medical Physics, Moscow Institute of Physics and Technology, National Research University, Dolgoprudny 141701, Russia; 2Department of Pathomorphology and Reproductive Toxicology, Research Center of Toxicology and Hygienic Regulation of Biopreparations, NRC Institute of Immunology FMBA of Russia, Ul. Lenina 102A, Serpukhov 142253, Russia

**Keywords:** injury, burn model, mesenchymal stem cell, hypoxia, cellular respiration

## Abstract

Transplantation of mesenchymal stromal cells (MSCs) provides a powerful tool for the management of multiple tissue injuries. However, poor survival of exogenous cells at the site of injury is a major complication that impairs MSC therapeutic efficacy. It has been found that tissue-oxygen adaptation or hypoxic pre-conditioning of MSCs could improve the healing process. Here, we investigated the effect of low oxygen tension on the regenerative potential of bone-marrow MSCs. It turned out that incubation of MSCs under a 5% oxygen atmosphere resulted in increased proliferative activity and enhanced expression of multiple cytokines and growth factors. Conditioned growth medium from low-oxygen-adapted MSCs modulated the pro-inflammatory activity of LPS-activated macrophages and stimulated tube formation by endotheliocytes to a much higher extent than conditioned medium from MSCs cultured in a 21% oxygen atmosphere. Moreover, we examined the regenerative potential of tissue-oxygen-adapted and normoxic MSCs in an alkali-burn injury model on mice. It has been revealed that tissue-oxygen adaptation of MSCs accelerated wound re-epithelialization and improved the tissue histology of the healed wounds in comparison with normoxic MSC-treated and non-treated wounds. Overall, this study suggests that MSC adaptation to ‘physiological hypoxia’ could be a promising approach for facilitating skin injuries, including chemical burns.

## 1. Introduction

Mesenchymal stromal cells (MSCs) play an important role in wound healing due to paracrine activity and exosome production that mediates replenishment of the stem cell pool at the site of injury, immunomodulation, ECM deposition, and angiogenesis [1]. The use of exogenous MSCs can accelerate tissue regeneration, although the low survival rate of transplanted cells in damaged tissue seems to be a significant limitation of MSC-based therapy [2,3,4,5]. Factors such as the release of pro-inflammatory mediators, ischemia, and mechanical and oxidative stresses in the microenvironment of the injured tissue are responsible for significant loss of the injected cells [3]. To address this issue, some strategies have been proposed, including genetic modifications [6,7,8], pre-conditioning with cytokines or growth factors [9,10,11], incorporation of the cells into scaffolds [12,13], and cultivation under a low-oxygen atmosphere [14,15,16,17,18,19,20,21,22]. Multiple studies have shown that hypoxic pre-conditioning (<2% O_2_) or adaptation to ‘physiological hypoxia’ (2–9% O_2_) may lead to an augmentation of the MSC therapeutic effect due to several mechanisms. First, decreased oxygen tension induces hypoxia-inducible factor 1 alpha (HIF1α) activation that leads to upregulation of *TP53*, *AKT*, and *BCL2* genes, facilitating proliferation and survival of MSCs [23,24]. Second, hypoxia enhances the production of immunosuppressive molecules, such as interleukin-10 (IL-10) and prostaglandin E2 (PGE2) [25,26], that limit the expansion of tissue damage. Moreover, low oxygen exposure stimulates pro-angiogenic and regenerative activity of MSCs via endothelial growth factor (VEGF) and hepatocyte growth factor (HGF) production [3,25,27]. It has been revealed that induction of autophagy in MSCs takes place under hypoxic conditions [28]. Upregulation of all mentioned molecules and signaling pathways occurs in a HIF1α-dependent manner. Such deep changes in the gene expression profile enable MSC adaptation and resistance to nutrient deprivation, oxidative stress, and pro-apoptotic factors within the injured tissue.

To date, the benefit of hypoxic pre-conditioning for MSC-mediated therapies has been demonstrated in animal models of acute kidney [17] and ischemic lung [16] injuries, myocardial infarction [29], traumatic spinal cord [15], and brain [18] injuries. Fewer studies have been carried out on the skin wound models. For instance, Muhammad et al. revealed a slight acceleration of acid burn wound closure on a mouse model after treatment with hypoxia (3% O_2_)-pre-conditioned adipose tissue-derived MSCs, although no functional characteristics of normoxic and hypoxia-pre-conditioned cells have been provided [22]. In the study by Yang et al., the authors achieved accelerated healing of diabetic wounds by application of hypoxia-pre-conditioned (1% O_2_ for 48 h) MSCs entrapped in dermal matrix as compared with matrix-treated wounds [21]. The limitation of the study is the lack of a group treated with matrix in combination with MSCs, grown under 21% O_2_. Another study reports about the involvement of hypoxia-mediated (5% O_2_ for 4 h) activation of PGC-1α/SIRT3 signaling in changing mitochondrial morphology and decreased level of cleaved caspase-3 in MSCs. The authors hypothesized that this signaling contributes to the survival of engrafted MSCs and, therefore, to a therapeutic effect on a model of cutaneous wound [20]. However, in this study, the influence of low oxygen tension on paracrine activity and MSC interaction with other cell types in skin wounds has not been explored.

Here, we aimed to investigate the influence of MSCs, cultivated under standard (21% O_2_) (hereafter, 21% oxygen MSCs or normoxic MSCs) and low-oxygen (5% O_2_) (hereafter, 5% oxygen MSCs) conditions, on the key players of tissue regeneration, including fibroblasts, endotheliocytes, and macrophages. Since MSCs cultured under hypoxic and normoxic conditions exhibit significant differences in cytokine and growth factor expression profiles, we next compared their tissue regenerative potential using a murine alkali burn injury model.

## 2. Results

### 2.1. Characterization of Bone Marrow MSCs

Phenotyping of bone marrow-derived MSCs has shown that isolated cells are positive for CD105, CD90, CD73, CD44, and CD29 (more than 90%) and negative for CD45, CD34, CD31, CD14, and CD11b (less than 5%) (Figure 1A). In addition, MSCs were able to differentiate into adipocytes, chondrocytes, and osteoblasts under the standard in vitro differentiation protocols (Figure 1B). Thus, the resulting MSC cell population has sufficient purity and meets the criteria established by the International Society for Stem Cell Therapy [30].

### 2.2. Influence of Oxygen Content on MSC Proliferation and Paracrine Activity

Once paracrine activity was believed to have a major contribution to MSC therapeutic efficacy, we compared the expression of biologically active proteins in tissue-oxygen-adapted and normoxic MSCs. Quantitative PCR analysis has shown that MSCs express a broad range of cytokines (IFNγ, IL6, TNFα, TGFβ1, IL10, IL1β, IL23A, and IL12) and growth factors (VEGFA, bFGF, and EGF) that participate in the wound healing process. We found a significant augmentation of their expression in MSCs cultured under a 5% oxygen atmosphere in comparison with MSCs cultured under a 21% oxygen atmosphere (Figure 2A). It was also found that tissue-oxygen-adapted MSCs exhibit increased levels of VEGFA expression on a protein level as well (Figure 2B). We also revealed an insignificant increase in VEGFA concentration in 5% oxygen MSC-conditioned growth medium (CGM) (Figure 2C). Besides cytokines, 5% oxygen MSCs show increased production of the inflammatory mediator prostaglandin E2 (PGE2) (Figure 2D), which also exhibits immunosuppressive effects such as inhibition of macrophage and T cell activation and contributes to the resolution of inflammation [31].

Once the expression level of growth factors influences cell viability and proliferation rate, we evaluated the mitotic activity of tissue-oxygen-adapted and normoxic MSCs by flow cytometry analysis of DNA profiles. It turned out that the 5% oxygen MSC population contains a higher number of cells in the S- and G_2_/M-phases of the cell cycle (Figure 2E). The colony formation assay has also shown an increased number and size of 5% oxygen MSC colonies as compared with their normoxic counterparts (Figure 2F,G) indicating enhanced proliferative activity of tissue-oxygen-adapted MSCs. The increased proliferative and secretory activity of MSCs grown under conditions of low oxygen content may contribute to the better survival and efficiency of cells during transplantation.

### 2.3. Impact of MSC-Conditioned Medium on LPS-Activated Macrophages, Fibroblasts, and Endotheliocytes

MSCs actively participate in the wound healing process via the production of regulatory signals and the recruitment of additional functional cells, such as macrophages, fibroblasts, and endotheliocytes [1]. Skin tissue injury is always accompanied by inflammation due to the release of damage-associated molecular patterns (DAMPs) from the necrotic cells. To study the in vitro interaction of 5% and 21% oxygen MSCs with activated macrophages, we used a LPS-activated macrophage assay. As shown by flow cytometry (Figure 3A), treatment with LPS resulted in 85% TNFα-positive macrophages in the RAW264.7 cell population.

It turned out that treatment of activated RAW264.7 macrophages with 5% and 21% oxygen in MSC CGM induced a 3-fold decrease in the percentage of TNFα-positive cells as compared with non-treated LPS-activated cells (Figure 3A). Noteworthy, in terms of TNFα expression level, determined as mean fluorescence per cell, treatment of activated RAW264.7 macrophages with 5% oxygen MSC CGM led to a more significant inhibition of TNFα expression than CGM from normoxic MSCs (Figure 3B).

Quantitative RT-PCR analysis has also shown a drop in TNFα expression after RAW264.7 macrophage treatment with MSC-derived CGMs at the transcriptional level (Figure 3C). Overall, treatment with 21% oxygen MSC CGM promoted inhibition of paracrine activity in LPS-activated macrophages, whereas the influence of 5% oxygen MSC CGM on macrophage behavior was more complex. For example, CGM from tissue-oxygen-adapted MSCs inhibited the expression of TNFα, whereas expression of many cytokines and growth factors almost remained unchanged (Figure 3C). At the same time, we observed a significant upregulation of IFNγ, IL1β, and EGF.

Migration of fibroblasts into the wound bed, followed by proliferation, significantly contributes to the granulation tissue formation at the proliferation stage of the wound healing. Using a scratch assay, we compared the influence of 5% and 21% oxygen MSC CGMs on fibroblast motility. It was found that CGM from tissue-oxygen-adapted MSCs stimulated “wound” closure to a much higher extent as compared with 21% oxygen MSC CGM (Figure 4A,B).

Production of pro-angiogenic molecules is a driving force of neoangiogenesis in the wound microenvironment and contributes to granulation tissue formation. MSC CGM contains VEGFA, which activates endotheliocytes and promotes vessel sprouting. The tube formation assay has shown that treatment of HUVEC monolayer with 5% oxygen MSC CGM results in the formation of a 2D network of tubes with an increased loop number and negligible amount of apoptotic cells as compared with serum-free medium or 21% oxygen MSC CGM (Figure 4C,D).

### 2.4. Therapeutic Effect of Tissue-Oxygen-Adapted MSCs in an Alkali Burn Injury Model

To evaluate the therapeutic effects of tissue-oxygen-adapted and normoxic MSCs, we selected an alkali burn injury model. The early changes in the chemical burn wound morphology were evaluated on day 6 after the injury. It turned out that the chemical burn caused a full-skin injury and affected the underlying *panniculus carnosus* muscle (Appendix A). We found that migrating epithelial tongues were formed in the wounds of all experimental groups, indicating an ongoing re-epithelialization process (Appendix A).

For the analysis of the therapeutic effect of MSC, we evaluated several parameters of animal recovery. Healing of the alkali burn wounds in all groups was mediated by re-epithelialization and wound contraction. Desquamation of primary eschar in 5% oxygen MSC-treated burn wounds has taken place in all animals before day 23 (Appendix A), whereas in other groups it has occurred later. It was found that the kinetics of wound closure in the 5% oxygen MSC-treated group was statistically significantly faster than in the saline-treated group (Figure 5A,B). No statistically significant differences were revealed between the 21% oxygen MSC-treated group and the saline-treated group. On day 28, primary and secondary eschars were still present in some saline-treated and 21% oxygen MSC-treated wounds, whereas in 5% oxygen MSC-treated wounds, no remaining eschars were observed (Appendix A). Moreover, all 5% oxygen MSC-treated wounds were re-epithelialized on day 28, whereas animals with incomplete re-epithelialization in other groups were present (Appendix A).

For semiquantitative analysis of wound morphology, tissue specimens were subdivided into three groups according to a histological score. A portion of the specimens demonstrated the formation of a thin, continuous epidermis with a well-developed *stratum corneum*. Newly formed hair follicles were present within the dermal tissue area, whereas scar tissue was highly dense. Wounds with such morphological properties were referred to as “grade 1” or “completely repaired.” The second group, or “grade 2” wounds, demonstrated “moderate repair.” These specimens represented the formation of a thick epidermis with a continuous *stratum corneum* and without newly formed hair follicles in the scar area. “Grade 3” wounds exhibited a thick and discontinuous epidermal layer with residual eschar. Scar tissue exhibited less density and leukocyte infiltration as compared with other grades. These wounds were classified as “poorly repaired” (Figure 5C). It was found that four of seven animals had grade 1 wounds in the 5% oxygen MSC-treated group, whereas only two animals had a “completely repaired” wound in the saline-treated and 21% oxygen MSC-treated groups. In addition, no animals with grade 3 wounds were found in the 5% oxygen MSC-treated group (Figure 5D). Thus, treatment of alkali burns with 5% oxygen MSCs resulted in accelerated wound healing with higher efficacy.

## 3. Discussion

It has been shown that allogenic MSCs provide a good therapeutic option for the treatment of skin injuries of different natures, including cutaneous wounds [32], thermal burns [33], and contact frostbites [34,35]. However, there are not too many studies that show a benefit of hypoxic preconditioning of MSC for the treatment of skin injuries.

Our data clearly show an increased secretory activity of tissue-oxygen-adapted MSCs. As shown in Figure 2, MSCs can produce a wide range of biologically active molecules, including growth factors and pro-inflammatory and anti-inflammatory cytokines, indicating the high extent of their plasticity. Most likely, elevated expression of growth factors (particularly EGF) in 5% oxygen MSCs contributes a lot to their enhanced viability and proliferation rate (Figure 2D–F), acting in an autocrine manner. The key role of the EGF/EGFR/Erk/Akt axis in self-renewal of MSCs was also investigated in the earlier study [36]. The authors have determined an inhibitory effect of androgen receptor (AR) on EGF-mediated signaling and proposed the use of anti-AR agents in clinical applications to improve the efficiency of MSC transplantation in treating various diseases. Clearly, tissue-oxygen adaptation seems to be a safer approach to improving MSC self-renewal. It was also shown that EGFR stimulation also promotes the production of angiogenic molecules in MSCs [37]. We also observed upregulation of VEGFA and bFGF in tissue-oxygen-adapted MSCs (Figure 2A–C) that can be attributed to EGF or HIF1α-mediated signaling. As a result, conditioned growth medium from 5% oxygen MSCs promoted an increased pro-angiogenic effect in vitro (Figure 4C,D). It was also shown that conditioned growth medium from tissue-oxygen-adapted MSCs significantly enhanced fibroblast motility (Figure 4A,B) as compared with CGM from normoxic MSCs, presumably due to upregulated bFGF, TGFβ1, and IL6 (Figure 2A) [38]. It should be noted that fibroblast migration and angiogenesis play an important role in the healing of skin injuries.

Alkali burns cause serious skin damage owing to lipid saponification and protein hydrolysis that lead to “liquefaction necrosis” [39]. As a result, the release of DAMPs from necrotic cells induces a strong inflammatory response, while the loss of skin barrier function makes the wound vulnerable to infections. We have shown that treatment of LPS-activated macrophages with conditioned growth medium from 5% and 21% oxygen MSCs decreased the percentage of TNFα-positive macrophages and TNFα expression level (Figure 3). TNFα is a cytokine that can promote apoptosis and lead to injury propagation at the inflammation stage of the wound healing process. TNFα blockade enables to minimize injury expansion as was shown on different animal models [40,41,42,43]. A favorable effect of anti-TNFα treatment has also been shown in a model of alkali corneal burn in mice [44]. It means that MSC-mediated inhibition of TNFα could significantly contribute to accelerated wound healing. In contrast to 21% oxygen MSCs, tissue-oxygen-adapted counterparts caused IFNγ upregulation in LPS-activated macrophages (Figure 3C) that could increase the resistance of the wounds to infections. Finally, conditioned growth medium from 5% oxygen MSCs significantly increased EGF and IL1β expression in activated macrophages (Figure 3C). IL1β mediates proliferation of epidermal keratinocytes and acts synergistically with EGF [45,46], resulting in accelerated wound closure. RAW264.7 macrophages treated with conditioned medium from normoxic MSCs did not show elevated expression of EGF and IL1β (Figure 3C). Most likely, this fact, along with the decreased self-renewal rate, resulted in fewer therapeutic in vivo effects of normoxic MSCs in the alkali burn model as compared with tissue-oxygen-adapted MSCs.

Local injection of 5% oxygen MSCs resulted in not only accelerated re-epithelialization but also improved quality of tissue recovery (Figure 5), presumably via the mechanisms mentioned above. In addition to paracrine activity, the increased therapeutic effect of 5% oxygen MSCs could be in part due to increased production of exosomes, which can also play an immunosuppressive and pro-angiogenic role [47]. It was found earlier that hypoxic pre-conditioning of MSCs led to upregulation of exosome secretion and enhanced expression of exosome biogenesis and secretion markers [48]. Although we did not evaluate the survival rate of the engrafted MSCs, obtained therapeutic effects clearly indicate an active involvement of the transplanted cells inthe healing process. It means that tissue-oxygen adaptation of MSCs is a simple and efficient method to improve their therapeutic properties for the treatment of skin injuries.

## 4. Materials and Methods

### 4.1. Isolation of Primary MSCs from Bone Marrow and Collection of MSC CGMs

Mouse bone marrow MSCs were isolated from the whole bone marrow aspirates of 4-week-old male Balb/c mouse (Stolbovaya, Russia). Briefly, MSCs were flushed from the bone marrow cavities of femurs and tibias with DMEM (Gibco, Grand Island, NY, USA), supplemented with 10% fetal bovine serum (Gibco, Grand Island, NY USA) and NEAA (Capricorn Scientific, Ebsdorfergrund, Germany). The cells were cultured in the same medium in a humidified atmosphere at 37 °C. Primary lines were cultured under normoxic (5% CO_2_, 21% O_2_) (21% oxygen MSCs) or hypoxic (5% CO_2_, 5% O_2_) (5% oxygen MSCs) atmospheres. The replacement of the growth medium was carried out daily during the first 4 days after isolation of the cells. Adherent proliferating cells were subcultured using 0.05 % Trypsin-EDTA (Gibco, Grand Island, NY, USA).

For CGM collection, 5% oxygen MSCs and 21% oxygen MSCs were seeded into 75 cm^2^ flasks. When a confluency of 30% has been reached, the growth medium was replaced with serum-free medium. After 72 h, CGM were collected, filtered through 0.22 µm PES filters, and stored at 2–8 °C.

### 4.2. Immunophenotyping of the Isolated MSCs

Primary MSCs at passage 3 were washed and stained with 100 μL solutions of antibodies against CD105 (120408, BioLegend, San Diego, CA, USA), CD73 (127220, BioLegend, San Diego, CA, USA), CD90 (206105, BioLegend, San Diego, CA, USA), CD29 (102205, BioLegend, San Diego, CA, USA), CD44 (203906, BioLegend, San Diego, CA, USA), CD31 (ab 28364, Abcam, Cambridge, UK), CD11b (ab25533, Abcam, Cambridge, UK), CD14 (ab182032, Abcam, Cambridge, UK), CD45 (103107, BioLegend, San Diego, CA, USA), CD34 (152204, BioLegend, San Diego, CA, USA), containing 0.5% FBS for 30 min at 4 °C in the dark. After staining, the cell suspensions were washed twice and analyzed using flow cytometer, CytoFLEX (Beckman Coulter, Brea, CA, USA). Forward and side scattering were used to exclude debris and dead cells from consideration. For each sample, more than 10,000 events were gated.

### 4.3. Differentiation of the Isolated MSCs

Primary MSCs at passage 3 were seeded on tissue culture 6-well plates. After reaching 80% confluency, the culture medium was replaced with a medium for differentiation from StemPro^®^ Adipogenesis, Osteogenesis, or Chondrogenesis Differentiation Kits (all from Gibco, Grand Island, NY). The cells were cultured according to the manufacturer’s protocol for 14 or 21 days. Then, differentiated and control undifferentiated MSCs were stained with Sudan III, Alizarin Red S, and Safranin O (all from Sigma-Aldrich, St. Louis, MO, USA), respectively. Cell images were obtained using DMi8 inverted microscope (Leica Microsystems, Wetzlar, Germany).

### 4.4. DNA Profile Analysis

The cells were removed from the flask, fixed in 66% ethanol solution, and stained with a solution of 1.67 μg per mL propidium iodide (Thermo Fisher, Grand Island, NY, USA) with 50 U per mL RNAse A (Thermo Fisher, Bremen, Germany) in PBS (Gibco, Grand Island, NY) for 30 min at 37 °C. Then, the cells were subjected flow cytometry analysis using CytoFLEX (Beckman Coulter, Brea, CA, USA).

### 4.5. Colony Formation Assay

MSCs were seeded into 6-well plates at low density (500 cells per well) and grown for 14 days under normoxic or hypoxic conditions with growth medium replacement once every three days. On day 14, the medium was aspirated, and the colonies were stained and fixed with 0.3% crystal violet solution (Ruskhim, Moscow, Russia) in ethanol at room temperature for 5 min. Then, the cells were thoroughly washed with water. The number of colonies was determined by photographing followed by processing using the ImageJ software package (1.42v, US National Institutes of Health, Bethesda, MD, USA).

### 4.6. Western Blot

The lysates of 5% and 21% oxygen MSCs with equal amount of protein (40 μg) were mixed with loading dye, boiled for 5 min, separated in denaturing 12.5% SDS-polyacrylamide gel, and transferred to Amersham^TM^ Hybond^TM^ 0.45μm PVDF membrane (GE Healthcare, Chalfont St Giles, UK). The membrane was blocked with 3% dry milk in TBS buffer with 0.1% Tween (TBS-T) for 1 h and incubated for 3 h at room temperature with antibodies against VEGFA (ab1316, Abcam, Cambridge, UK) and β-actin (ab8227, Abcam, Cambridge, UK) as a control at recommended dilutions. Then, the membranes were washed twice with TBS-T and incubated with HRP-conjugated secondary antibodies (ab205719 and ab205718, Abcam, Cambridge, UK) at room temperature for 1 h, followed by several washings with TBS-T and deionized water. Protein bands were visualized by ChemiDoc XRS+ imaging system (Bio-Rad Laboratories, Hercules, CA, USA) using chemiluminescence mode.

### 4.7. Immunofluorescent Analysis

The amount of PGE2 in 5% and 21% oxygen MSC lysates was determined using Prostaglandin E2 ELISA Kit (ab133021, Abcam, Cambridge, UK) according to the manufacturer’s instructions. The obtained values were normalized to the concentration of total protein in the lysate measured using BCA Protein Assay Kit (Pierce Biotechnology, Rockford, IL, USA) according to the manufacturer’s instructions.

### 4.8. BLItz Analysis of VEGFA Content

The concentration of VEGFA in MSC CGMs was determined using bio-layer interferometer BLItz Pro (ForteBio, Pall Life Sciences, NY, USA) as described earlier [27]. Briefly, antibodies to VEGFA (ab52917, Abcam, Cambridge, UK) were immobilized on biosensors with conjugated protein A. After washing in PBS, the biosensors were incubated in serum-free growth medium to set a new baseline, followed by the measurement of MSC CGM experimental sample (Appendix A). The VEGFA concentration in the sample was calculated from the linear calibration, which was determined using the standards with concentrations of 0.1, 1, 10, 100, and 1000 ng VEGFA per mL.

### 4.9. Flow Cytometry Analysis of TNFα Expression in LPS-Activated Macrophages after the Treatment with MSC CGM

RAW264.7 macrophage cells (ATCC TIB-71) were cultured in DMEM supplemented with 10% (*v*/*v*) FBS and 1% penicillin/streptomycin. The cultures were maintained at 37°C in a humidified 5% CO_2_ atmosphere. For LPS stimulation, RAW264.7 cells were seeded in a 6-well plate at a density of 200,000 cells per well and cultured up to 70–80% confluency. Then, the cells were washed three times with HBSS, followed by 30 min treatment with LPS at a concentration of 100 ng per mL in serum-free medium or with serum-free medium alone as a negative control. Subsequently, the LPS-activated cells were thoroughly washed with HBSS and incubated with 5% and 21% oxygen MSC CGMs for 24 h. The cells without activation were cultured in the same non-conditioned medium.

Next, the cells were washed three times with cold PBS, harvested, and fixed in 0.01% paraformaldehyde for 10–15 min at 4 °C. Afterwards, the cells were permeabilized with 0.2% Tween in PBS for 10 min at room temperature, washed and blocked with 10% FBS in PBS for 20 min, and stained with primary anti-TNFα antibody (ab215188, Abcam, Cambridge, UK) and secondary antibody conjugated with Alexa 647 (ab269823, Abcam, Cambridge, UK). After the staining, the cells were washed three times with PBS to remove unbound secondary antibodies and resuspended in PBS for flow cytometry analysis using CytoFLEX (Beckman Coulter, Brea, CA, USA).

### 4.10. RT-PCR

RT-PCR was used to determine molecular expression profiles of MSCs and LPS-activated RAW264.7 cells after the treatment with 5% and 21% oxygen MSC CGMs. The cells were lysed using Extract RNA reagent (Evrogen, Moscow, Russia) according to the manufacturer’s recommendations. Purified RNA samples were subjected to spectral analysis. If the absorption ratio A260/A280 in at least one TRIZOL-purified sample was lower than 2.0, the samples were repurified using CleanRNA Standard (Evrogen, Moscow, Russia). The integrity of the purified RNA was assessed by electrophoresis in 1.5% agarose gel under non-denaturing conditions. The obtained RNA samples were converted to cDNA using the MMLV RT kit (Evrogen, Moscow, Russia). These samples were subjected to qPCR with the primers listed in Appendix A using CFX96 Real-Time PCR Detection System (Bio-Rad, Hercules, CA, USA). The 18S RNA gene was used as an endogenous control. The results were analyzed with the CFX Manager software supplied by the manufacturer. All samples were run in triplicate. Overall, three independent experiments were performed.

### 4.11. Tube Formation Assay

Human umbilical vein endothelial cells (HUVECs) were obtained from the cell culture collection of the Koltzov Institute of Developmental Biology of the Russian Academy of Sciences and cultured in medium 199 with Earle’s salts supplemented with 10% fetal bovine serum (FBS), sodium pyruvate, ITS, hydrocortisone (0.5 μg per mL), bFGF (2.5 ng per mL), EGF (10 ng per mL), heparin (15 IU per mL), HEPES, and penicillin/streptomycin. The cells were incubated at 37 °C in a humidified atmosphere containing 5% CO_2_. HUVECs (passage 6). For fluorescent labeling, the cells were seeded onto T25 flask, followed by a treatment with CellTracker™ Green CMFDA dye (Invitrogen). Then, they were harvested using 0.25% trypsin solution and seeded at a density of 200,000 cells per well on 24-well plate coated with collagen I (Biolot, St. Petersburg, Russia). After 24 h, the culture medium was removed, and 200 μL of collagen/Matrigel mixture (20 µL 10× PBS, 53 µL Collagen I gel (Corning, Somerville, MA, USA), 1.5 µL 1 N NaOH, 125.8 µL Matrigel per well to achieve a final collagen I concentration of 1 mg per mL) were added as a layer above the cells and allowed to gelate for 30 min at 37 °C. Fresh growth medium (control) or 5% and 21% oxygen MSC CGMs were added after the Collagen/Matrigel mixture had gelled, and tube formation was observed after 6 h. The number of loops was determined by analysis of images obtained using fluorescent microscope AxioVert.A1 (Zeiss, Oberkochen, Germany). Three random fields were examined for each group.

### 4.12. Scratch Assay

NIH/3T3 fibroblasts (ATCC CRL-1658), cultured in DMEM supplemented with 10% (*v*/*v*) FBS and 1% penicillin/streptomycin, were used for cell migration assay as described earlier [49]. In detail, 200,000 cells per well were seeded on 6-well plates and cultured until reaching a monolayer. Then, the cell monolayers were scratched using a sterile 200 µL plastic tip in the middle of each well and gently washed with HBSS to remove the detached cells and debris. Next, 5% and 21% oxygen MSC CGMs and non-conditioned medium were added separately. At 0, 12, 24, 36, and 48 h after the wounding, the scratched regions were photographed using AxioVert.A1 microscope (Zeiss, Oberkochen, Germany) equipped with a Plan 10×/0.25 phase-contrast objective. Image J software (NIH, Bethesda, MD, USA) was used to quantify the cell migration.

### 4.13. In Vivo Study

Thirty 8-week-old male Balb/c mice (Stolbovaya, Russia) with an average weight of 18–24 g were used in this study. The mice had access to food and water ad libitum and were housed at room temperature. The work with animals was carried out in accordance with the international guidelines for the care and use of laboratory animals. The study protocol was approved by the Bioethics Commission of the Research Center of Toxicology and Hygienic Regulation of Biopreparations.

A chemical burn was caused under anesthesia by a 3 min contact of the depilated skin on the back with 40% aqueous solution of potassium hydroxide, resulting in fourth-degree burn wounds of 10 mm in diameter. After 24 h, the animals were randomly subdivided into 3 groups of 10 animals each. The first group of mice was locally injected with 1 million 5% oxygen MSCs in 200 µL of saline; the second group received the same amount of 21% oxygen MSCs, whereas the third (control) group was injected with 200 µL of saline. The surface area of the wound was measured on days 7, 14, 23, and 27 post-injury using the program Universal Desktop Ruler V2.8.1110 software (AVPSoft, Pittsburgh, PA, USA). Wound tissues were collected from 3 mice in each group on day 6, and from 7 mice on day 28 after burn injury. The excised samples were placed in 10% formalin solution for fixation. Then, the tissues were dehydrated and embedded in paraffin, cut into 5–6 μm-thick sections, and stained with hematoxylin and eosin. Analysis was performed using a microscope, Olympus CX41 (Olympus Co., Tokyo, Japan), equipped with a UPlanApo 20×/NA 0.70 objective lens.

### 4.14. Statistics

All experiments were performed at least three times. Exact number of samples or animals is specified in figure captions. The data are expressed as the mean ± standard deviation. The statistical significance between the specific group and control was determined using SPSS 13.0 software (IBM Corps., Armonk, NY, USA). Statistical evaluation between two groups was performed using a Mann–Whitney U-test. *p* < 0.05 was considered to indicate a statistically significant difference.

## 5. Conclusions

In this study, we first showed a beneficial effect of tissue-oxygen-adapted MSCs for alkali burn treatment in comparison with cells cultured under standard conditions. The improved therapeutic effect of tissue-oxygen-adapted MSCs seems to be a result of an enhanced self-renewal rate and elevated paracrine activity. We showed that 5% oxygen MSCs interact with key participants of the wound microenvironment, including fibroblasts, macrophages, and endothelial cells. In particular, tissue-oxygen-adapted MSCs stimulated fibroblast migration capacity and endotheliocyte-mediated tube formation to a much higher extent than their normoxic counterparts. Moreover, 5% oxygen MSCs decreased TNFα expression in activated macrophages while upregulating the expression of EGF and IL1β, which could significantly contribute to accelerated re-epithelialization. Evaluation of tissue-oxygen-adapted MSCs’ therapeutic effect in an alkali burn model has shown that these cells demonstrated faster wound closure as compared with normoxic MSCs. It should be noted that the treatment with 5% oxygen MSCs also improved skin tissue repair and hair follicle structure recovery. Thus, tissue-oxygen adaptation is a simple and efficient method to improve the therapeutic properties of MSCs.

## Figures and Tables

**Figure 1 ijms-24-04102-f001:**
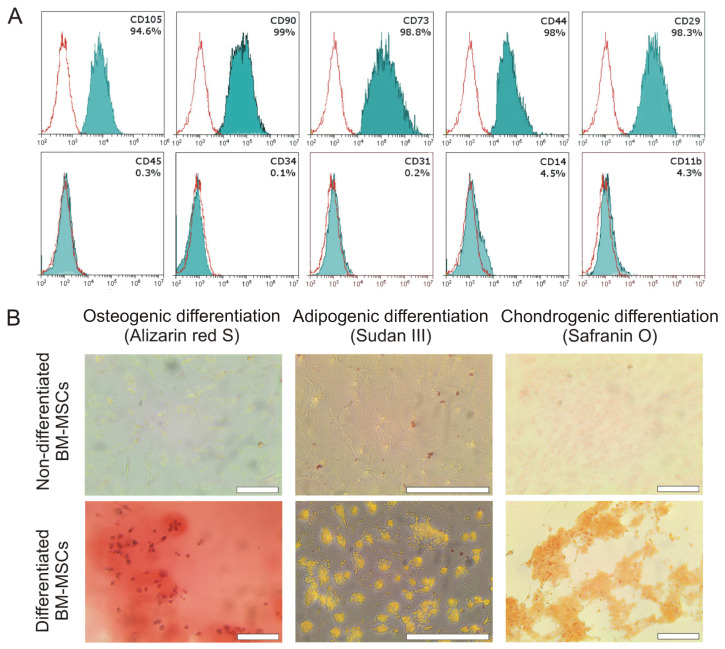
Characterization of isolated mouse MSCs. (**A**) Phenotyping of MSCs indicated more than 90% purity of the cell population, measured by flow cytometry. (**B**) The isolated MSCs were able to differentiate into osteoblasts, chondrocytes, and adipocytes under differentiation conditions (upper panel) that were determined by the staining with Alizarin Red S, Safranin O, and Sudan III, respectively. Scale bar is 200 µm.

**Figure 2 ijms-24-04102-f002:**
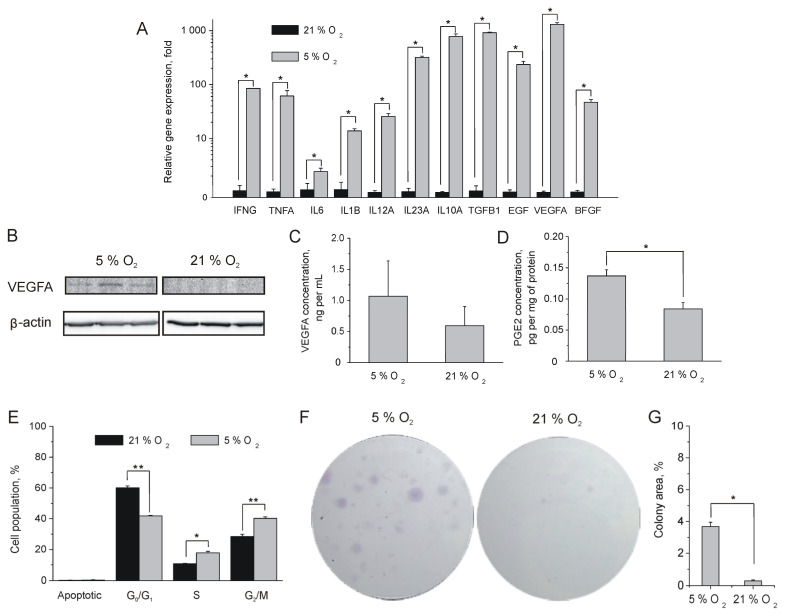
Paracrine activity and proliferation rate of MSCs. (**A**) Quantitative RT-PCR analysis of MSCs continuously cultured under 5% and 21% oxygen atmospheres. (**B**) Western blot analysis of VEGFA expression in tissue-oxygen-adapted and normoxic MSCs. (**C**) Comparison of VEGFA content in the MSC-conditioned media after 72 h in a 5% and 21% oxygen atmosphere determined by BLI analysis. (**D**) Immunofluorescent analysis of PGE2 production in tissue-oxygen-adapted and normoxic MSCs. (**E**) Quantitative assay of cell fractions at different cell cycle phases in MSC populations cultured under 5% and 21% oxygen atmospheres. (**F**) Colony-formation assay of MSC proliferation rate. (**G**) Comparison of tissue-oxygen-adapted and normoxic MSC colony areas on day 21. * *p* < 0.05, ** *p* < 0.01, Mann–Whitney U test. (**A**,**C**,**D**,**E**,**G**) Data are shown as means ± SD (*n* = 3).

**Figure 3 ijms-24-04102-f003:**
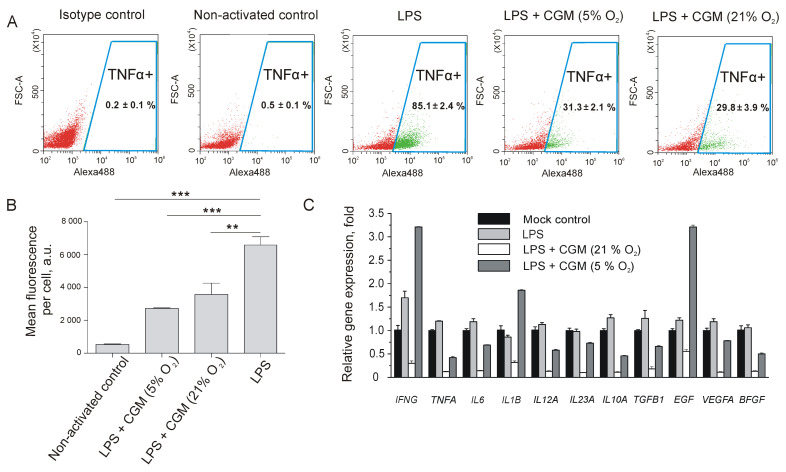
Influence of conditioned growth medium from MSC on LPS-activated RAW264.7 macrophages. (**A**) Images of TNFα-positive fractions of LPS-activated macrophages treated with conditioned growth media (CGM). (**B**) Influence of MSC CGM treatment on TNFα expression level in RAW264.7 macrophages determined as mean fluorescence per cell using flow cytometry. ** *p* < 0.01, *** *p* < 0.001, Mann–Whitney U test. (**C**) Quantitative RT-PCR analysis of RAW264.7 macrophages continuously cultured under 5% and 21% oxygen atmospheres. (**A**–**C**) Data are shown as means ± SD (*n* = 3).

**Figure 4 ijms-24-04102-f004:**
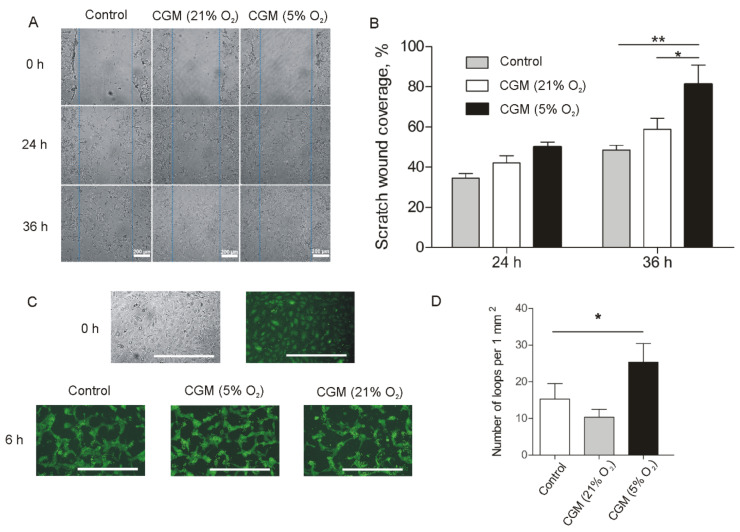
Influence of conditioned medium from MSC on fibroblasts and endotheliocytes. (**A**) Representative time-lapse images of 3T3 fibroblast scratch assays immediately after the scratches had been made and then after 24 h and 36 h in the presence of serum-free MSC CGM versus control medium. (**B**) There was a significant increase in the extent of wound closure in 3T3 fibroblast scratch assays in 5% oxygen MSC CGM compared with the control medium and 21% oxygen MSC CGM at 36 h. (**C**) Representative time-lapse images of CellTracker Green-stained HUVECs before treatment and then after 6 h incubation with serum-free growth medium (control), 5%, and 21% oxygen MSC CGMs. Scale bar is 200 µm. (**D**) Quantitative analysis of loop formation. (**B**,**D**) Data are shown as means ± SD (*n* = 3). * *p* < 0.05, ** *p* < 0.01, Mann–Whitney U test.

**Figure 5 ijms-24-04102-f005:**
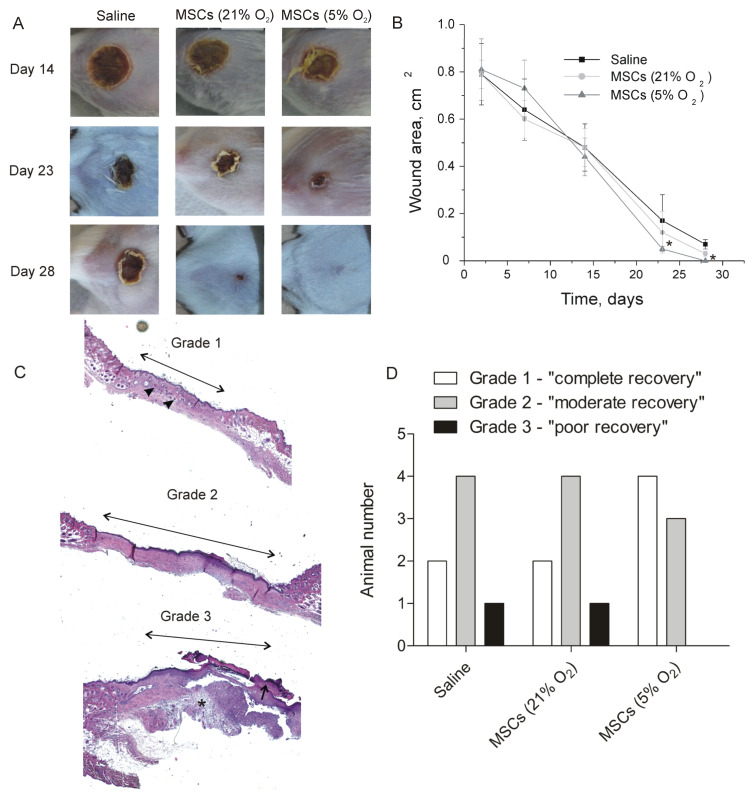
Wound healing in alkali burn model after treatment with saline, 5%, or 21% oxygen MSCs. (**A**) Representative images of the treated wounds taken post-injury on days 14, 23, and 28. (**B**) Treatment of the wounds with MSCs led to accelerated wound closure kinetics. The data are expressed as the mean ± standard deviation (*n* = 7). * *p* < 0.05, Mann–Whitney U test. (**C**) Representative images of the wounds of different histological scores. Grade 1 wounds exhibited thin stratum corneum and newly formed hair follicles (black arrowheads) in the scar area (double-headed arrow). Grade 2 wounds demonstrated thick continuous epidermis in the scar area (double-headed arrow) without hair follicles. Grade 3 wounds exhibited discontinuous and thick epidermis in the scar area (double-headed arrow) with a residual eschar (black arrow) and leukocyte infiltration (black star). (**D**) Distribution of the saline- and MSC-treated wounds according to histological score (*n* = 7).

## Data Availability

The data presented in this study are available on request from the corresponding author.

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
