# Peer review of "Tissue-Oxygen-Adaptation of Bone Marrow-Derived Mesenchymal Stromal Cells Enhances Their Immunomodulatory and Pro-Angiogenic Capacity, Resulting in Accelerated Healing of Chemical Burns"

_ijms, 2023, doi:10.3390/ijms24044102_

Round 1
Reviewer 1 Report
Please see the minor comments:
1. Line 37 should be rewritten as Factors such as....
2. Figure 1 panel B can be more clear and legible.
Line 138: 3-fold increase compared to what? Please clarify.
Line 140: How does this line correspond to the figure 3B plot? Maybe you need to provide information on what bar is normoxic MSCs?
The remaining parts and figures look well organized and well written.
Author Response
Reviewer #1
Please see the minor comments:
- Line 37 should be rewritten as Factors such as....
Re: The sentence was edited in the updated version of the manuscript.
- Figure 1 panel B can be more clear and legible.
Re: We added the labels about differentiation direction to the top part of the panel.
Line 138: 3-fold increase compared to what? Please clarify.
Re: We specified it (page 4, lines 139-141): It turned out that treatment of activated RAW264.7 macrophages with 5 % and 21 % oxygen MSC CGM induced 3-fold decrease in percentage of TNFα-positive cells as compared with non-treated LPS-activated cells (Figure 3A).
Line 140: How does this line correspond to the figure 3B plot? Maybe you need to provide information on what bar is normoxic MSCs?
Re: In Introduction we provided a definition of normoxic MSCs (lines 73-75): “Here, we aimed to investigate the influence of MSCs, cultivated under the standard (21 % O2) (hereafter, 21 % oxygen MSCs or normoxic MSCs) and low-oxygen (5 % O2) (hereafter, 5 % oxygen MSCs) conditions, […]”
The remaining parts and figures look well organized and well written.
Re: Thank you for a positive review!
Reviewer 2 Report
Authors have compared the therapeutic efficacy of BM-MSC grown under normoxia and hypoxia via in vitro assays as well as animal model of burn. The manuscript written well and easy to read. Below are my comments/questions.
Fig 1-A: The 55 Oxygen condition, up-regulated an array of potent pro-inflammatory genes (ITNFa, L1B, IL12A and IL23A) that could exert a detrimental effect in many disorders and exacerbate the injury. How could we evaluate the net effect of the measured cytokines.
Fig 2B: What was the rational to only choose do choose VEGF-A for protein analysis. The Western blot bands are not convincing, there is a faint band only with one the replicates under 5% oxygen and no band with 21% oxygen.
Fig 2C: The difference between two groups seems to be statistically insignificant (Lack of asterisks on the graph). If so, the write up in the result section should be edited to note this. Overall, a cytokine array analysis of conditioned medium would be more informative.
Fig 3C. Although authored showed both CGM-5% and CGM-21% decreased TNFa positive macrophages (Fig 3A-B), but the gene expression panel on Fig 3C showed that CGM-5% failed to inhibit pro-inflammatory phenotype of macrophages. Except for TNFa, other pro-inflammatory markers remained unchanged or even increased. Although, pro-inflammatory macrophages may be beneficial to prevent infection in early phase of burn, but their persistence may be detrimental. Could authors explain based on these results and Fig 1-A data, how MSC grown under hypoxia could be more beneficial than normoxia.
Authors showed a therapeutic efficacy of MSC treatment in rodent burn model, however, they didn’t conduct any experiments to explore the mechanism of action of hypoxic MSC. In discussion section, they speculated that one mechanism could be through reduction of TNF-a positive macrophages, however, in Fig1A, they showed that Hypoxic MSC express almost 100 times more TNFa compared to normoxia that makes the conclusion difficult. They also proposed a mechanism via induction of EGF which is possible, but they didnt provide any experiment to support this.An experiment with EGF-silenced MSC could be useful to address this and enhance the quality of the manuscript for publication in IJMS.
Author Response
Reviewer #2.
Authors have compared the therapeutic efficacy of BM-MSC grown under normoxia and hypoxia via in vitro assays as well as animal model of burn. The manuscript written well and easy to read. Below are my comments/questions.
Fig 1-A: The 55 Oxygen condition, up-regulated an array of potent pro-inflammatory genes (ITNFa, L1B, IL12A and IL23A) that could exert a detrimental effect in many disorders and exacerbate the injury. How could we evaluate the net effect of the measured cytokines.
Re: We did not aim to evaluate the net effect of the measured cytokines because besides them many immune regulatory cytokines and growth factors were also upregulated in 5% oxygen MSCs. Therapeutic experiment using murine alkali burn injury model has shown that local injection of 5% oxygen MSCs does not exacerbate the injury. Conversely, treatment with 5% oxygen MSCs accelerated wound re-epithelialization and improved tissue histology of the healed wounds in comparison with normoxic MSC-treated and non-treated wounds.
Fig 2B: What was the rational to only choose do choose VEGF-A for protein analysis. The Western blot bands are not convincing, there is a faint band only with one the replicates under 5% oxygen and no band with 21% oxygen.
Re: We chose VEGF-A for protein analysis because of the important role of this growth factor, which mediates neoangiogenesis during the proliferative stage of wound healing. In turn, neoangiogenesis significantly contributes to wound tissue supply with oxygen and nutrients as well as to recruitment of immune cells participating in healing process. Another cause of our interest in analysis of VEGF-A expression is its dependence on hypoxia level that was mentioned in Discussion (lines 244-247). We also increased contrast in WB image (Fig. 2B) to make it more convincing.
Fig 2C: The difference between two groups seems to be statistically insignificant (Lack of asterisks on the graph). If so, the write up in the result section should be edited to note this. Overall, a cytokine array analysis of conditioned medium would be more informative.
Re: Yes, the difference between two groups in Fig.2C is statistically insignificant that can be attributed to relatively high growth medium volume and therefore high dilution (SD for the method increases with dilution). We re-wrote it in a more correct manner (line 104): “We also revealed insignificant increase in VEGFA concentration in 5 % oxygen MSC conditioned growth medium (CGM) (Figure 2C).”
Fig 3C. Although authored showed both CGM-5% and CGM-21% decreased TNFa positive macrophages (Fig 3A-B), but the gene expression panel on Fig 3C showed that CGM-5% failed to inhibit pro-inflammatory phenotype of macrophages. Except for TNFa, other pro-inflammatory markers remained unchanged or even increased. Although, pro-inflammatory macrophages may be beneficial to prevent infection in early phase of burn, but their persistence may be detrimental. Could authors explain based on these results and Fig 1-A data, how MSC grown under hypoxia could be more beneficial than normoxia.
Re: We agree that 5% oxygen MSC CGM only in part modulated pro-inflammatory macrophage activity mediating drop in TNFα expression (Fig.3). However, it increased EGF and IL1β expression, which accelerate re-epithelialization process during wound healing (refs. 45,46). Besides macrophages, tissue oxygen-adapted MSCs exhibited upregulated VEGF-A and bFGF levels (Fig.2A) and stimulated endotheliocytes and in vitro angiogenesis (Fig.4C,D). It was also found that conditioned growth medium from tissue oxygen adapted MSCs significantly enhanced fibroblast motility (Fig.4A,B) as compared with CGM from normoxic MSCs presumably due to upregulated bFGF, TGFβ1 and IL6 (Fig.2A). Thus, increased expression of some cytokines and growth factors in 5% oxygen MSCs could stimulate regenerative properties of several participants in wound healing process. We discussed it in the text (lines 244-272).
Authors showed a therapeutic efficacy of MSC treatment in rodent burn model, however, they didn’t conduct any experiments to explore the mechanism of action of hypoxic MSC. In discussion section, they speculated that one mechanism could be through reduction of TNF-a positive macrophages, however, in Fig1A, they showed that Hypoxic MSC express almost 100 times more TNFa compared to normoxia that makes the conclusion difficult. They also proposed a mechanism via induction of EGF which is possible, but they didnt provide any experiment to support this. An experiment with EGF-silenced MSC could be useful to address this and enhance the quality of the manuscript for publication in IJMS.
Re: As we mentioned above, MCSs are not a single player in wound healing microenvironment. Engrafted MSCs exert their effects via paracrine interaction with different types of cells resulting in their activation (or reprogramming) and alteration in expression profile. We experimentally showed that MSCs induce 3-fold decrease in percentage and expression level of TNFα-expressing LPS-activated macrophages (Fig.3A,B). Moreover, 5% oxygen MSCs induced significant EGF upregulation in LPS-activated macrophages (Fig.3C). It shows that MSCs contribute to macrophage reprogramming. Second, 5% oxygen MSCs exhibited increased VEGF-A production in comparison with normoxic MSCs (Fig.2A-C) that resulted in increased tube formation by endotheliocytes (Fig.4C,D). Third, we experimentally showed that CGM from tissue oxygen adapted MSCs stimulated fibroblast migration to a much higher extent as compared with 21 % oxygen MSC CGM (Fig.4A,B). We found that 5% oxygen MSCs exhibit increased expression of IL6 (Fig.2A), which modulates fibroblast migration as was established earlier (10.1111/j.1365-2133.2007.07867.x). We suppose that the mentioned links between expression level of biologically active molecules in MSCs, their content in CGMs, effects of CGMs on the behavior of different cell types and identifying of CGM-mediated alterations in macrophage expression profiles provide the rationale of wound healing mechanisms of tissue oxygen-adapted MSCs.
I would like to thank the Reviewers again for their contribution in making the manuscript a better and more lucid evocation of the work as a result of their suggestions and criticisms. I thank you very much for your recommendations and assistance.
Round 2
Reviewer 2 Report
I would like to thank the authors for addressing my comments/questions either by editing the manuscript or by providing justifications.